# A Prospective Observational Study to Determine the Efficacy of a Theatre Prioritisation Tool in Optimal Utilisation of Limited Theatre Time for Deep Burn Injury in a Resource-Restricted Setting

**DOI:** 10.3390/ebj6040055

**Published:** 2025-10-17

**Authors:** Nikki Leigh Allorto, Reitze Rodseth, David Gray Bishop

**Affiliations:** 1Pietermaritzburg Metropolitan Department of Surgery, University of KwaZulu-Natal, 201 Townbush Road, Pietermaritzburg 3201, South Africa; 2Netcare Ltd., Johannesburg 2196, South Africa; reitzerodseth@gmail.com; 3Department of Anaesthesiology and Critical Care, Nelson R. Mandela School of Medicine, University of KwaZulu-Natal, Durban 4001, South Africa; 4Perioperative Research Group, Department of Anaesthetics, Critical Care and Pain Management, University of KwaZulu-Natal, 201 Town Bush Road, Pietermaritzburg 3201, South Africa; davidgbishop@gmail.com

**Keywords:** burn injury, surgery, low resource setting, sepsis, treatment strategy, triage

## Abstract

**Background:** Routine early surgery for all deep burns in low-resource settings is not currently achievable. We designed and implemented a simple triage strategy that selected patients to be prioritised for early surgery based on a more urgent need and greater potential benefit. The primary outcome was the ability to perform surgery in the priority group within three days of the decision. **Methods:** This was a prospective, descriptive study undertaken at a tertiary hospital in Pietermaritzburg, South Africa. All patients referred to the Grey’s Hospital Burn Service were triaged into either priority or non-priority groups. Priority designation was based on total burn surface area (TBSA) > 15%, the presence of sepsis, or limb-threatening injury. Data related to demographic information, injury, and outcomes were collected and managed using REDCap electronic data capture tools. **Results:** There were 191 admissions with 42 (22%) meeting priority criteria. The priority group had larger burns (TBSA 25 [Interquartile range 15–30] vs. 8 [3–15]%) and included all septic injuries. We provided early surgery within a median of 1.4 (interquartile range 0.5–3.3) days of the decision for surgery being made. A total of 75% of patients were operated within 72 h of the decision, and 43% within 10 days of injury. The system identified a sicker cohort, as evidenced by high mortality, ICU admission, and acute kidney injury rates. In the non-priority group, reported outcomes were more positive, but with a high injury-to-discharge days per percentage TBSA. **Conclusions:** This simple triage strategy represents a novel approach for prioritising access to burn surgery in a setting where global surgery standards are desirable but not always possible. We were able to identify the high-risk groups and provide surgery within acceptable time frames. Future research should be aimed at refining this triage system and improving outcomes in the priority group.

## 1. Introduction

The high burden of burn injury in low- and middle-income (LMIC) settings is well described [1,2]. Effective management is compromised by resource restriction, with access to theatre a key limitation. Deep burn injury requires excision and grafting. This represents definitive care that attenuates physiological derangements, infection risk, and restores skin integrity. The global standard of care for early surgery is 7 to 10 days post-injury [3]. This paradigm developed in the 1970s after Janžekovič’s work, with clear mortality benefit demonstrated in early surgery [4]. While these results have not been replicated in low-resource settings when early surgery is performed, it remains the global standard of care [5,6]. In our burn service, a retrospective study showed that only 16% of patients received surgery within 7 days and 27% within 10 days of the burn injuries [7].

Aiming to provide routine *early* surgery in all deep burns in low-resource settings is not currently achievable. Access to theatre is a limited resource and unlikely to improve in our setting, given increasing fiscal restraints. The need for triage where the burden of injury outweighs available resources is well described in mass casualty literature [8,9,10]. The acute mass influx of patients temporarily overwhelms system resources, and management of these mass casualties relies heavily on the ability to access other facilities and specialist personnel in the region. The triage system focuses on burn assessment and referral to the appropriate facility and facilitation of emergency surgery. Total body surface area [TBSA], age, and inhalation injury are considered to be the main predictors of resource utilisation and outcomes. A key report by Saffle and colleagues presents triage tables in order to provide an objective determination for resource allocation for improved quality of decision making [11,12]. The challenge in our local setting, and many LMICs, is the persistent lack of adequate resources relative to the burden of care. There is little redundancy in the system and no other facility or specialist personnel that can be accessed in the manner that the burn assessment teams are in mass casualty incidents. There is also a distinct shortage of theatre time and access to the intensive care unit. No simple, objective system has been described in the literature that assists with the daily triage and prioritisation of surgery for burns in low-resource settings on a continual basis. Mass casualty literature also describes burn assessment teams, which can be used for specialised in-hospital triage, or specialised secondary triage, of patients who have already been admitted to a hospital. We therefore sought to utilise this concept and design and implement a simple, reproducible strategy that can be employed in similar settings where the health system is unable to meet the demand. This systematic approach aimed to select patients to be prioritised for early surgery based on a more urgent need and greater potential benefit for the patient. This concept is well described by the Institute of Medicine on crisis standards of care, which is defined as a substantial change in the usual health care operations and the level of care it is possible to deliver, justified by specific circumstances and formally declared by a state government in recognition that crisis operations will be in effect for a sustained period [13]. In contrast, the climate of health care in low-resource settings, such as that described for burn injuries in South Africa, has not been a change in care that was previously delivered. Rather, the crisis has been ongoing, and we are attempting to provide a framework for management of care delivery.

This study aimed to describe the efficacy of this prioritisation system. The primary outcome was the ability to perform surgery in the priority group within three days of the decision. We also report the time from injury to operation and aimed to describe the outcomes of both the priority and non-priority groups. This study was not designed to compare outcomes between priority and non-priority groups.

## 2. Materials and Methods

This was a prospective, descriptive study undertaken at Grey’s Hospital, a tertiary hospital in Pietermaritzburg, KwaZulu-Natal, South Africa. The Pietermaritzburg Burn Service (PBS) consists of a regional hospital (specialist support from the tertiary hospital is provided) and a tertiary hospital with an onsite specialist burn surgeon and 12 dedicated burn beds. All referrals to the PBS are made using the Vula medical referral application directly to the burn specialist [14]. There is only one dedicated burn theatre list per week at the tertiary hospital. Access to additional theatre time is restricted to utilising the emergency board, which is shared by all surgical specialties and caters to an average of 80 emergency surgeries per week. This board is prioritised according to the urgency of the surgery.

During the study period, all patients referred to the Grey’s Hospital burn service were triaged by the single burn specialist into either priority or non-priority groups.

The criteria for priority surgery were any one of the following:Larger surface area burns (deep burns > 15% TBSA for physiological source control);Invasive burn wound infection as part of sepsis source control;Full thickness with fourth degree components of the hand, foot, or limb burn with high risk of limb loss or significant deformity or dysfunction if not operated on early.

The non-priority group included deep injuries less than 15% total body surface area, not involving the hands, and where the limb burns had no fourth-degree component.

We originally designed the study with the TBSA group including 15–30% TBSA, because patients with more than 30% TBSA traditionally have poor outcomes in our setting. However, we did treat some patients with injuries larger than 30% TBSA. We undertook curative management in these cases and therefore included them in the analysis, given that they impacted the system resources. Priority designation was performed on referral and reassessed daily. Priority cases were given preference for transfer and admission to Grey’s Hospital as well as for surgery on the weekly elective list. To expand the operating time, we utilised the emergency operating theatre for priority cases. The same surgeon performed all surgeries, but a variety of nursing and anaesthetic teams were utilised, with staff who do not necessarily meet criteria for burns expertise as described in the mass casualty literature. Non-priority patients were accommodated in the next available bed and theatre list. Data related to demographic information, injury and referral details, time and date of decision for priority, date of surgery, and reasons for delay and outcomes (ICU admission, acute kidney injury, graft loss, and mortality) were collected and managed using REDCap electronic data capture tools hosted at the University of KwaZulu Natal [15]. Inhalation injury was diagnosed using bronchoscopy within 72 h of injury. Acute kidney injury was marked as present if any of the KDIGO (Kidney Disease: Improving Global Outcomes) criteria were present at any time during admission. Infection is a difficult diagnosis to make and was always made by the burn specialist surgeon in conjunction with an intensivist, and not by more junior doctors. Sepsis was always carefully considered, and diagnosis was based on clinical evaluation according to international guidelines, blood and X-ray investigation, and cultures. Antibiotics were initiated for suspected sepsis [16,17]. To reduce inter-observer variability, the same surgeon and the same intensivist made the decision to start antibiotics using the PBS burn service guideline (Appendix A). Graft loss was marked present if more the 5% of the graft surface area failed, as assessed by the single burn specialist. Ethical approval is given by the University of KwaZulu Natal (BREC 106/14) as class approval for the burns database, and specifically for this prospective observational study (BREC/00004082/2022).

### Statistics

The sample size was based on convenience sampling over an 18-month period. Data analysis was conducted using REDCap electronic data capture tools hosted at the University of KwaZulu Natal. Baseline characteristics of the included patients were reported as mean (standard deviation [SD]) for continuous normally distributed variables; median (interquartile range [IQR]) for data not normally distributed; and count (percent) for categorical variables.

To address the primary endpoint, we reported the time from designation as a priority patient to surgery, as well as the proportion of patients who received surgery within 72 h of this decision. Secondary outcomes included the proportion of prioritised patients who received surgery within 24 and 48 h of decision, as well as the proportion of patients who received surgery within 3, 7, and 10 days from the time of injury. We further reported mortality, length of stay, time from injury to Greys hospital discharge, ICU admission, septic episodes, acute kidney injury, graft loss, and factors preventing access to surgery.

We used the Strengthening the Reporting of Observational Studies in Epidemiology (STROBE) statement guidelines to report our study [18]. The STROBE checklist is attached as Appendix A.

## 3. Results

Data collection occurred from 1 January 2022 to 30 June 2023. The flow chart for the overall cohort is presented in Figure 1. There were 191 admissions to the tertiary hospital over the 18-month period. Median patient age was 3.5 years old (IQR 1–35), with 56% being children under 12 years, and 54% male. Median TBSA was 12% (IQR 5–24), with the mechanism being hot water in 44%, flame in 31%, low voltage electrical injury in 8%, high voltage in 5% and the remainder including hot food, hot oil, hot surface, and chemical burns. Inhalation injury was present in 6%. Thirty-three (17%) patients were non-operative admissions. The non-operative group included patients with larger TBSA burns that were superficial partial thickness, where surgery was not indicated (26 patients), or patients with deep burns that deteriorated before the decision for surgery was made and palliative care was instituted (7 patients).

The patient groups are described in Table 1, linked to length of stay outcomes. A total of 42 patients (22% of admissions) met priority criteria, with 49 priority procedures performed.

The primary outcome per group is provided in Table 2, together with injury-related time periods. Median time from injury to surgery was seven days. For the primary outcome, once the decision for priority was made, the time to surgery was a median of 1.4 days (IQR 0.5–3.3). The time from injury to decision was a median of 6.5 days (IQR 3.4–10.5). Time from decision to surgery was within 24 h in 43% of surgeries, within 48 h in 68%, and within 72 h in 75%. Time from injury to surgery was within three days for 13% of operations, seven days for 38% and 10 days for 43%.

Reasons for delays in surgery are presented in Table 3. The most common cause of delayed priority surgery was theatre availability, with preference given to other surgical cases deemed more urgent; followed by the patient being too sick requiring further resuscitation, and the availability of the surgeon.

Secondary outcomes are presented in Table 4. The rate of negative outcomes in the priority group was high, with 30% mortality, 55% incidence of acute kidney injury, and 40% admission to the ICU. A total hospital stay was 1.3 days per % TBSA. No amputations were performed in the priority group. In the sepsis group, resolution of sepsis was not a specific data point that we collected, and is thus not reported.

If surgery was staged, adherence to the current protocol was maintained for all operations. Eleven patients had two operations, three patients had two operations, and one patient had four operations. For simplicity, we have presented data on the first surgery only. Sixty-one percent of admissions did not fall into the priority group and demonstrated a long time to surgery of 33 days and a 4.7 total days hospital stay per % TBSA. However, this group had better outcomes with 2.6% mortality, ICU admission in 4.3%, acute kidney injury in 11% and 32% with graft loss.

## 4. Discussion

This study aimed to assess the effectiveness and impact of a triage system that was developed to optimise the use of limited theatre time in a resource-restricted environment where time to surgery was the primary outcome measure. The main findings were that 22% of patients met priority criteria, and we were able to provide early surgery within 1.4 days of the decision for surgery being made in this group. This simple system successfully identified a higher-risk group, evidenced by outcomes such as higher mortality, ICU admission, and acute kidney injury rates. In the non-priority group, reported outcomes were more positive, but with a high injury-to-discharge days per % TBSA.

The priority criteria were based on accepted burn care principles. For example, surgery is a cornerstone of sepsis management to eliminate the source and control ongoing contamination [19]. Where there is an invasive burn wound infection or a high burden of non-viable tissue with risk of invasive wound infection, prompt surgery is warranted. In larger surface area deep burns, early surgery is considered part of physiological source control. TBSA is well described as the predominant predictor of outcome together with age and inhalation injury, particularly in the austere and mass casualty setting [11,12]. Inhalation injury is uncommon in our setting, mostly seen in association with massive TBSA injury, and where palliative care would be provided. Where inhalation injury is associated with a manageable TBSA burn, therapeutic intervention is attempted. Full-thickness burns of the hand, foot, or limb burn were selected when there was a high risk of limb loss or significant deformity or dysfunction if not operated on early. Obvious full-thickness facial burns are typically associated with very large surface area injuries, which in our setting are treated with palliative care. Typically, facial burns are partial thickness or mixed depth and managed expectantly and grafted at 3 weeks when demarcation is clear. Similarly with perineal burns. Most of our patients are children and young adults, with older patients being uncommon and generally having poor outcomes, so they were not included in priority criteria due to poor likelihood of benefit and lack of geriatric specialists. Our selective strategy contrasts with the standard approach, where the aim is to operate on *all* patients requiring surgery within 7 to 10 days of injury [3] on a first-come come first-served basis. We deliberately delayed the transfer of non-priority patients from district-level care to accommodate priority cases. Attempts were then made to provide surgery for the prioritised patients as soon as the decision for priority was made, by giving these patients preference on available burns lists and on the emergency surgery lists.

The standard of care in high-income countries for deep burns is early surgery. Poorer outcomes have been demonstrated in low/middle-income countries when early surgery is performed compared to delayed surgery [5,20]. This may suggest that not all surgery should be performed within ten days in this setting; however, a subgroup might benefit from early surgery. Therefore, there is a need for a system that identifies this subgroup and enables early surgery in these settings. We used a strategy of triage, previously described in mass casualty incidents, that aimed to systematically identify patients for early surgery in a setting where early surgery cannot be universally provided. We utilised principles described in the guidelines for burn care under austere conditions, including adoption of staged procedures and surgical pace matched to the level of other team members when the emergency theatre list was utilised [21]. We sought to develop and implement a simple method for patient selection, and we evaluated the strategy by time to surgery for the prioritised group. While the outcomes were worse in this prioritised group, this was not unexpected, as it represents a subgroup of patients with bigger burns or with active sepsis. Further work will be required to identify areas that may be targeted to lower these negative outcomes, including an assessment of the optimal timing of surgery in our setting.

Authors from The Netherlands have investigated the timing of burn surgery. Dokter et al. found that the time to surgery was 14.7 days post-burn in the early 2000s [22]. More recently, the Dutch Burn Repository Group published a multicenter cohort study investigating the timing of surgery in acute burn care [23]. They conducted a retrospective review of 3335 adult patients between 2009 and 2021, who underwent a surgical procedure. They found that 20% of patients received surgery within 7 days of injury, with a median of 4 days (IQR 1–6) in the early group and 16 days (IQR 12–20) in the late group, where early surgery is defined as 7 days. These times are comparable to both our current study and our retrospective data [7]. They report older age, larger total body surface area burned, psychiatric disorders, and mechanism of flame, flash, or contact burns to be predictors of early surgery with worse outcomes in the early surgery group (mortality of 7.5%, sepsis 6.7% and partial graft take 14.2%), which they attribute to the higher severity of injury in the early group.

Our findings were similar, with worse outcomes in the patients receiving earlier surgery. However, the length of hospital stay was shorter (1.3 versus 4.7 days per % TBSA) in our study. This contrasts with the Dutch group, who report 23 versus 9 days LOS in the early and late groups. Length of stay data should ideally be reported as days per % TBSA. When converting the Dutch data with median TBSA of 10% and 3%, the result of 2.3- and 3 days LOS per % TBSA is more comparable to ours and is one of the benefits of early surgery.

Our results show that the triage strategy correctly identified the high-risk groups. We were able to provide surgery for three-quarters of these patients within 72 h of decision, and 43% received surgery according to international guidelines of 10 days from injury. There was an acceptable time from injury to discharge, but with a high complication and mortality rate. This is in keeping with the recent publication of the Dutch Burn Repository Group.

The higher mortality rate in the priority group (12/42 [29%]) versus the non-priority group (3/112 [2.6%]) is concerning. This may be due to the priority designation identifying patients with increased severity of illness. However, this study was not primarily designed to answer this question, and we advise that these results be taken with caution. Our findings are consistent with a publication from Malawi in 2015, demonstrating higher mortality [25.3% vs. 9.2% (*p* = 0.001)] in the early surgery group, where early surgery was defined as within 5 days of injury [17]. Further prospective studies are needed to address this concern definitively, and may impact the surgical approach to the timing of burn surgery required in LMICs.

Further scrutiny of each priority group may have value and reveal areas that could be targeted for improvement. The TBSA group consisted mostly of children sustaining hot water burns with good outcomes, despite all patients having some graft loss. This is expected in large surface area grafts. One of the goals of early excision in large surface area burns is the prevention of invasive burn wound infection, which we have achieved in this group. Once sepsis developed, the outcomes were worse and with a higher rate of intensive care admission. This group also had a 12% incidence of inhalation injury, which is known to worsen outcomes [24,25]. However, we feel it was the development of sepsis that was the critical predictor of outcome. In the group with large surface area scalds, the initial focus of management is resuscitation and wound care to prevent depth conversion and infection. Surgical management is expectant, as we only consider surgery for deep burns. It is not our practice to take superficial partial thickness burns for excision. We wait for a clear demarcation of depth rather than operate on burns that will potentially heal spontaneously. Once the depth or presence of infection is clear, surgery becomes a priority. This is illustrated by our data, with the longest time from injury to decision of 13.6 days. Part of the challenge in this group is the presentation of the patient to a district-level hospital, frequently run by junior doctors with little burn knowledge and skill. The initial critical resuscitation is often performed inadequately, even with expert advice, and the transfer to our hospital is often delayed and may hamper effective resuscitation. Wound care in that setting is usually suboptimal. This aspect could be a target for intervention in future studies.

The worst outcomes were seen in the group with sepsis and TBSA less than 15%. This reflects a high-risk subcategory, accounting for 6% of the total admitted injuries with TBSA < 15%. The outcome is difficult to explain, with this group receiving the shortest time to surgery (3.5 days).

### 4.1. Strengths and Weaknesses

Due to the burn service being managed by a single specialist, there was consistent application of the system and diagnosis of infection, with reduced inter-individual bias. However, the system requires external validation through further studies to assess performance in the hands of different health care workers. We believe the system is simple to understand and easy to use, and that our context is broadly generalisable to other LMIC settings.

We have established a clear protocol for the rationing of scarce resources that is objective, transparent, and easily implemented in contexts where experienced burn care practitioners are not available. This is a starting point that we believe should be audited and refined in order to provide early surgery to the group of patients most likely to benefit. It is based on need and likelihood of survival and promotes transparency, fairness, and prevention of discrimination. In terms of ethical considerations, we have addressed fairness, duty to care, resource stewardship, transparency, consistency, proportionality, and accountability according to the core ethical principles guiding decision making in Crisis Standards of Care [13]. We believe that this strategy is generalizable and can be adopted in other low-resource contexts, given that limited access to theatre is a common problem in LMICs. The profile of burn injuries should be assessed together with evaluation of the local barriers to care provision (for example, theatre access, ICU access, blood products). Criteria applicable to the local case mix can be developed with a view to prioritising access for those most likely to benefit, rather than access on a first-come, first-served basis.

Our study did not include data on long-term outcomes, such as hypertrophic scarring and contracture, which would be expected in patients with delays from the time of injury to surgery. The lack of data on the long-term outcomes of scarring and function is an important limitation. There may also have been unavoidable selection bias: patients referred to Grey’s Hospital with an initial non-priority classification may have changed condition, and re-discussion is dependent on the referring doctor. There are potentially some patients who may have met priority criteria but were never referred or not referred again when the condition changed. While we believe these numbers to be small due to the referral system being well established since 2017, these omissions are difficult to quantify.

### 4.2. Future Areas of Research

Further studies should investigate if early surgery is appropriate in the low-income setting, specifically whether the need for early surgery selects a sicker group of patients or the implementation of early surgery results in poorer outcomes.

We identified a high-risk group with smaller surface area injuries that had poor outcomes despite early surgery. Future studies should investigate predictors of poor outcomes within this group of TBSA injury < 15% and consider targeted interventions to improve outcomes. Our criteria for prioritisation seem to align with international approaches, but the criteria could be refined further, and this should be investigated. In the group with large TBSA injury that developed sepsis, the early period of care at district level hospitals could be targeted for improvement. Overall, we should investigate how outcomes can be improved in the group prioritised for early surgery; considerations include improvement of perioperative care. Long-term data on scarring should be sought.

## 5. Conclusions

Global surgery standards are desirable but not always possible. This paper describes a novel adaptation of the system for surgical access in a low-resource setting faced with challenges in universal access to early surgery for all deep burns. We used a strategy of triage, previously described in mass casualty incidents, to improve access to early surgery for the patients we believed would most benefit. We were able to identify the high-risk groups using a simple system and provide surgery within acceptable time frames. While the outcomes were poorer compared to the non-priority group, who were managed more conservatively, this group represented a sicker sub-population. Future research should be aimed at refining this triage system and improving outcomes in the priority group.

## Figures and Tables

**Figure 1 ebj-06-00055-f001:**
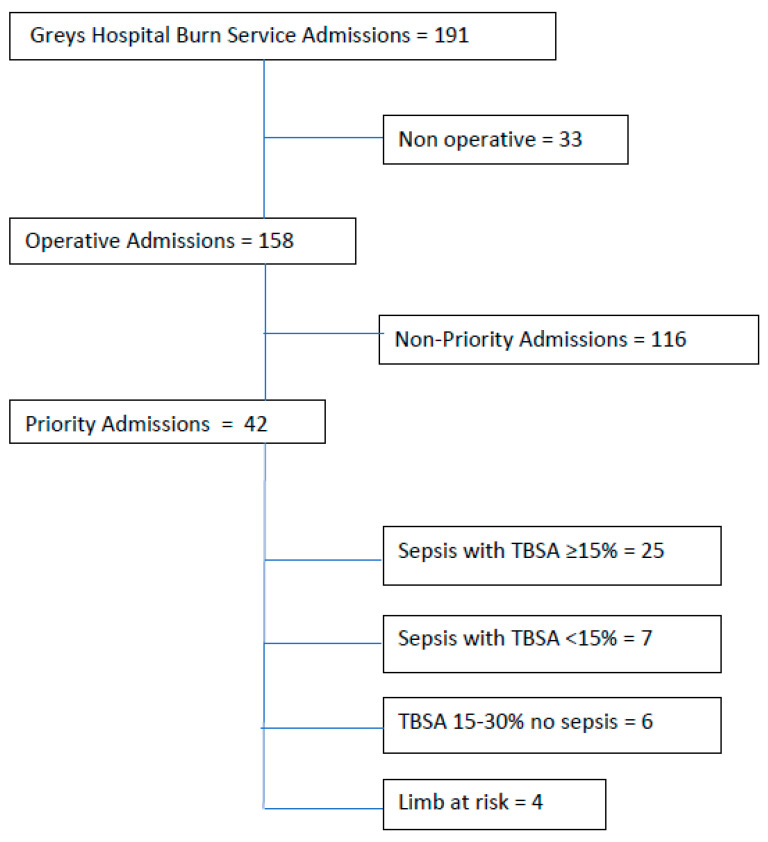
Patient pathway flow chart. TBSA, total body surface area.

**Table 1 ebj-06-00055-t001:** Description of patient groups (age, TBSA, and presence of inhalation injury) and length of stay data.

n	Group Cohort	Age	TBSA	Inhalation Injury	Days to Referral	Injury to Admission	LOS GH	Injury to Discharge
33	Non operative	2 [1–33]	17 [10–25]	7 (24%)	0	2 [1–10]	13 [7–16]	15 [10–31]
116	Non-priority operations	4.5 [1–34]	8 [3–15]	1 (0.9%)	3 [0–16]	19 [8–40]	17 [11–25]	38 [28–60]
42	Priority operations	2 [1–38]	25 [15–30]	3 (7%)	0 [0–1]	2 [1–5]	30 [15–48]	33 [16–51]
Priority Subcategories								
6	TBSA category (≥15%) No infection present	1 [0.3–1.6]	30 [30–32]	0	0 [0–5.3]	3.5 [1–6.75]	51 [43–56]	54 [49–62]
25	TBSA category (≥15%) Infection present	2 [2–31]	25 [16–34]	3 (12%)	0 [0–1]	2 [1–5]	33 [19–48]	36 [25–52]
7	Sepsis category (TBSA < 15%)	35 [16.5–58]	9 [6–12]	0	0 [0–0.5]	2 [1–2]	15 [8–17]	16 [11–19]
4	Limb at risk category	43.5 [37–51]	2.5 [1–5]	0	0	4 [2.5–6]	6 [4–16]	12 [7–23]

n, number of patients; TBSA, total body surface area; LOS GH, length of stay at Grey’s Hospital; ‘injury to discharge’ includes length of stay at referral hospital. Reported as median and [interquartile range] or number (percentage of group).

**Table 2 ebj-06-00055-t002:** Primary outcomes of priority surgeries include injury to decision, injury to surgery, and decision to surgery.

Surgery Category	Injury to Decision	Decision to Surgery	Injury to Surgery
TBSA category (≥15%) No infection present n = 9	13.6 [6.4–18.7]	3.6 [2.4–8.6]	22 [10–24]
TBSA category (≥15%) Infection present n = 29	6.4 [3.7–9.6]	1.3 [0.5–1.6]	7 [5–13]
Sepsis category (TBSA < 15%) n = 7	2.6 [2–3.5]	0.9 [0.4–1.5]	3.5 [3–4]
Limb at risk category n = 4	2 [2–6.4]	1.4 [1.1–2.4]	5 [4.8–6.5]
All priorities n = 49	6.5 [3.4–10.5]	1.4 [0.5–3.3]	7 [4–14.5]
Non-priority operations n = 125	-	-	33 [21–55]

n, number of patients in the group; TBSA, total body surface area. Days reported as median and [interquartile range].

**Table 3 ebj-06-00055-t003:** Reasons for delayed access to theatre for prioritised surgery.

	Too Sick	No PICU Bed	Theatre Availability	No Surgeon	No Anaesthetist
First 24 h n = 34	13 (38.2%)	1 (2.9%)	14 (41.2%)	6 (17.7%)	0
Second 24 h n = 25	8 (32%)	1 (4%)	11 (42%)	5 (20%)	0
Third 24 h n = 16	4 (25%)	0	10 (62.5%)	2 (12.5%)	0
Fourth 24 h n = 14	3 (21.4%)	0	11 (78.6%)	0	0

n, number in the group (percentage); PICU—paediatric intensive care.

**Table 4 ebj-06-00055-t004:** Description of secondary patient outcomes, including ICU admission, infections, acute kidney injury, graft loss, mortality, and days per percentage LOS.

n	Group Cohort	ICU Admission	Number of Infection Episodes	Acute Kidney Injury	Graft Loss	Mortality	Days per % LOS
33	Non operative	11 (33%)	1 [1–2]	9 (27%)	–	9 (27%)	0.9
116	Non-priority operations	5 (4.3%)	0	13 (11%)	37 (32%)	3 (2.6%)	4.7
42	Priority operations	16 (40%)	2 [1–4]	23 (55%)	21 (50%)	12 (30%)	1.3
Priority Subcategories							
6	TBSA category (≥15%) No infection present	1 (17%)	4 [2.5–4]	1 (17%)	5 (83%)	0	1.8
25	TBSA category (≥15%) Infection present	13 (52%)	3 [1–4]	15 60%	10 (40%)	9 (36%)	1.4
7	Sepsis category (TBSA < 15%)	4 (57%)	1 [1–2]	6 (86%)	1 (14%)	5 (71%)	1.8
4	Limb at risk category	1 25%	0.5	1 (25%)	0	1 25%	4.8

n, number of patients in the group (percentage); TBSA, total body surface area; LOS, length of stay; ICU, intensive care unit; ICU admission, acute kidney injury and graft loss presented as percentage of total group, infection episodes as number of episodes and [interquartile range]; Days per percentage LOS = days from injury to Greys Hospital discharge per percentage TBSA burn.

## Data Availability

Data were collected and managed using REDCap electronic data capture tools hosted at the University of KwaZulu Natal and can be made available on request.

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
