# Peer review of "A Prospective Observational Study to Determine the Efficacy of a Theatre Prioritisation Tool in Optimal Utilisation of Limited Theatre Time for Deep Burn Injury in a Resource-Restricted Setting"

_2673-1991, 2025, doi:10.3390/ebj6040055_

Round 1
Reviewer 1 Report (New Reviewer)
Comments and Suggestions for Authors
This article reports on the efficacy of a triage system for burn surgery in a low middle income country. The authors were able to prioritize operations based on the system. The concept of development of a triage system is a worthy one that can influence care in austere environments. A few questions.
- Although the time from decision to operate and operation was 3 days, the time from injury to decision was 15 days. This defeats the purpose of shortening time to the OR. Can time to decision be decreased?
- Does this prioritization system make a difference compared to the standard practice? The paper would be powerful with a comparator group from the non-triage time.
- What were the outcomes? Were limbs salvaged in the limb threatened group? How many in the sepsis group had resolution of sepsis?
Author Response
Please see the attachment

Reviewer 2 Report (New Reviewer)
Comments and Suggestions for Authors
This is a valuable and timely study that addresses a very real challenge in burn care in resource-limited settings. The idea of adapting mass casualty triage principles to everyday practice is innovative and clinically relevant. The study is well designed, the prospective approach is a strength, and the results are presented clearly and systematically.
A few areas could be strengthened:
- Provide more detail on how sepsis was diagnosed and how inter-observer variability was minimized.
- Consider including a more explicit descriptive comparison between the priority and non-priority groups, which would help emphasize the value of the triage system.
- Expand the discussion on generalizability — how might this model be adapted to other low-resource contexts with different constraints?
- Acknowledge more explicitly the lack of long-term outcomes (e.g., scarring, function) as an important limitation.
- Improve the table legends slightly so that they can be understood without referring back to the text.
Overall, the manuscript is well written, original, and provides practical insights that will be of real interest to the readership.
Comments on the Quality of English LanguageThe English is fine and does not require major improvement. Minor grammatical and stylistic edits could further improve clarity, but the current level is acceptable.
Round 2
Reviewer 2 Report (New Reviewer)
Comments and Suggestions for Authors
Well solved and no comments
Comments on the Quality of English LanguageAceptable
This manuscript is a resubmission of an earlier submission. The following is a list of the peer review reports and author responses from that submission.
Round 1
Reviewer 1 Report
Comments and Suggestions for Authors
I read that and found it valuable. Although it has some corrections such as references' correction, that two of them are old.
this article needs to be corrected from references and English improvement.
Regards
Author Response
Comment 1
I read that and found it valuable. Although it has some corrections such as references' correction, that two of them are old.
this article needs to be corrected from references and English improvement.
Response 1
Thank you for the comments. We have updated the reference that refers to inhalation injury. The reference for Janžekovič is the seminal paper from 1970 which led to the paradigm shift in surgical approach that we are discussing and is still practiced today, so we believe it is still a relevant reference.
We have improved the English through a detailed review of the paper, in consultation with experienced scientific writers, and also appropriate software analysis.
Reviewer 2 Report
Comments and Suggestions for Authors
Dear colleagues!
The topic of the article is very relevant, although not new. The real conditions of providing care to burn victims, described in detail in the article, are undoubtedly interesting. The article not only touches upon an acute medical problem, but also clearly demonstrates the inequality in the conditions of providing emergency surgical care in countries around the world. I have no doubt that the details of the professional activities of colleagues from South Africa will be of interest not only to surgeons in the countries of the "global south", but also to their colleagues from the countries of the comparatively prosperous "global north".
And yet, the article needs correction. In its current form, it loses a significant part of its value due to some shortcomings.
- Section "Abstract". The list of keywords should be supplemented. Sepsis, treatment strategy, Burn mass casualt, Triage, Mass Casualty Incidents - these or other similar terms will make the article more "recognizable".
- Section "Introduction". The authors state, among other things, that "The literature does not describe a system that helps in prioritizing surgical interventions for burns in resource-poor settings" (line 53,54). I cannot agree with this statement. The literature contains quite a lot of results of private studies and international recommendations on this matter. Relative resource shortages arise not only due to the lack of resources in the surgical hospital (wards, surgeons, operating rooms, and similar elements of the hospital infrastructure). Resource shortages often arise even with a relatively developed infrastructure when it is faced with a mass influx of victims. In such conditions, it is always necessary to sort patients into groups depending on the severity of their condition and the need for emergency surgical interventions and intensive care. It is very important to describe in the introduction at least the main international recommendations on this matter (https://doi.org/10.1016/j.burns.2022.12.011; 10.1016/j.burns.2020.07.001; 10.1097/BCR.0b013e31829afe25). Even if the authors believe that these recommendations are not applicable to their work, it is necessary to analyze these recommendations. It would be very interesting to know the reasons why the authors consider these recommendations inapplicable to the working conditions of a specific burn center in South Africa. Please do this in the Introduction or Discussion section. 3. Section "Introduction", lines 57-63. Here the authors describe the criteria for dividing patients into groups, which the authors themselves defined and used in the work. It is better to place such data not in the Introduction section, but in the Material and Methods section. In the Introduction section, it would be more appropriate to discuss the criteria for identifying priority groups of patients that are traditionally used in such situations: total burn area, deep burn area, shock, patient age. Please describe here the reason why you decided to limit yourself to sepsis, burn area, and specific anatomical location of the burn - extremities. Do you think that the patient's age, facial and perineal burns should not be a criterion for priority surgical intervention?
- The goal of the work is formulated quite specifically. In my opinion, the goal sounds a little unusual. As a rule, the goal (and the concept of treatment effectiveness) include the task of reducing mortality, reducing the incidence of complications, and similar clinical priorities. But the authors certainly have the right to set such a "technical" goal - reducing the duration of the preoperative period by expanding access to the operating room.
- Section "Materials and Methods". Line 107 - "Infection was marked present if systemic antimicrobials were 107 initiated". The meaning of the phrase is unclear. Did you think that a patient had a systemic infection if the doctor prescribed antibiotics to the patient? Is that right? Perhaps it means that "patients diagnosed with a systemic infection were prescribed antibiotics by the doctor"? Perhaps this misunderstanding is due to an inaccurate translation, but the phrase requires clarification.
- In the Material and Methods section, it is absolutely necessary to describe how exactly the diagnosis of "sepsis" was established.
- The Results section is written quite well. However, it remains unclear how the developed system of differentiated access to the operating room affected the clinical outcomes of treatment. Has the mortality rate among patients with severe burns become higher than it was before the introduction of the new system? Remained the same? Decreased? Perhaps this result (reduced complication rates and mortality) was not the goal of this work. But it is definitely the main purpose of the surgeon's work. Are there any data on this?
- The Conclusions section begins with the phrase "This triage strategy represents a novel approach for access to burn surgery in a setting where global surgical standards are desirable but not always possible." I repeat, there are serious doubts that this is precisely a "novel approach." The criteria that the authors used to divide the patient flow are rather traditional. This, however, does not reduce the value of the presented article.
Thus, the presented article is very interesting, informative, relevant. At the same time, it requires some revision.
best regards
Author Response
General Comment
The topic of the article is very relevant, although not new. The real conditions of providing care to burn victims, described in detail in the article, are undoubtedly interesting. The article not only touches upon an acute medical problem, but also clearly demonstrates the inequality in the conditions of providing emergency surgical care in countries around the world. I have no doubt that the details of the professional activities of colleagues from South Africa will be of interest not only to surgeons in the countries of the "global south", but also to their colleagues from the countries of the comparatively prosperous "global north".
Response
Thank you for the time taken to review our paper. We appreciate the time and effort that has been given. The comments were valuable and we believe have helped us significantly improve the paper. Please see answers to each comment below, with changes to the manuscript are marked in red.
Comment 1
And yet, the article needs correction. In its current form, it loses a significant part of its value due to some shortcomings.
- Section "Abstract". The list of keywords should be supplemented. Sepsis, treatment strategy, Burn mass casualty, Triage, Mass Casualty Incidents - these or other similar terms will make the article more "recognizable".
Response 1
Thank you for the suggestion, additional keywords have been included. We have not included the ‘Mass casualty’ keywords, as our submission does not deal with this scenario, although there are similarities in response patterns.
Comment 2
- Section "Introduction". The authors state, among other things, that "The literature does not describe a system that helps in prioritizing surgical interventions for burns in resource-poor settings" (line 53,54). I cannot agree with this statement. The literature contains quite a lot of results of private studies and international recommendations on this matter. Relative resource shortages arise not only due to the lack of resources in the surgical hospital (wards, surgeons, operating rooms, and similar elements of the hospital infrastructure). Resource shortages often arise even with a relatively developed infrastructure when it is faced with a mass influx of victims. In such conditions, it is always necessary to sort patients into groups depending on the severity of their condition and the need for emergency surgical interventions and intensive care. It is very important to describe in the introduction at least the main international recommendations on this matter (https://doi.org/10.1016/j.burns.2022.12.011; 10.1016/j.burns.2020.07.001; 10.1097/BCR.0b013e31829afe25). Even if the authors believe that these recommendations are not applicable to their work, it is necessary to analyze these recommendations. It would be very interesting to know the reasons why the authors consider these recommendations inapplicable to the working conditions of a specific burn center in South Africa. Please do this in the Introduction or Discussion section. 3. Section "Introduction", lines 57-63. Here the authors describe the criteria for dividing patients into groups, which the authors themselves defined and used in the work. It is better to place such data not in the Introduction section, but in the Material and Methods section. In the Introduction section, it would be more appropriate to discuss the criteria for identifying priority groups of patients that are traditionally used in such situations: total burn area, deep burn area, shock, patient age. Please describe here the reason why you decided to limit yourself to sepsis, burn area, and specific anatomical location of the burn - extremities. Do you think that the patient's age, facial and perineal burns should not be a criterion for priority surgical
intervention?
Response 2
Thank you for this comment. We agree that there are parallels between the mass casualty scenario in ‘developed’ settings and the setting we describe, as both deal with triage principles when resources are overwhelmed. A key difference is that emergency mobilisation of resources in mass casualty scenarios relates to a short-term scenario and includes the mobilisation of resources from facilities that may have additional capacity, while our setting deals with a chronic resource shortage that does not have additional resources available. However, the principles of prioritisation remain relevant, which the mass casualty literature does cover extensively. Our setting requires a system for prioritising which patients that are already within the hospital should gain access to theatre, when there are limited theatre slots available. This requires we prioritise operative management in a sub-group most likely to benefit from time-sensitive management. This needs to occur on a long-term basis, as there are no additional resources to mobilise. Our system specifically aims to create a simple, objective manner in which to choose these priority patients, acknowledging that fewer than 1 in 6 will receive operative management within ten days of burn injury. We have not found a system dealing with this specific aspect in the existing literature.
We have revised the introduction to incorporate the mass casualty literature and better describe the differences with our setting. We have also moved the description of the patient groups into materials and methods as suggested. We have also included the rationale for the selection of our criteria in our setting, which we expanded upon in the discussion.
Aiming to provide routine early surgery in all deep burns in low-resource settings is not currently achievable. Access to theatre is a limited resource and unlikely to improve in our setting, given increasing fiscal restraints. The need for triage where the burden of injury outweighs available resources is well described in mass casualty literature[8-10]. The acute mass influx of patients temporarily overwhelms system resources, and management of these mass casualties relies heavily on the ability to access other facilities and specialist personnel in the region. The triage system focuses on burn assessment and referral to the appropriate facility and facilitation of emergency surgery. Total body surface area [TBSA], age and inhalation injury are considered to be the main predictors of resource utilization and outcomes. The challenge in our local setting, and many LMICs, is the persistent lack of adequate resources relative to the burden of care. There is little redundancy in the system and no other facility or specialist personnel that can be accessed in the manner that the burn assessment teams are in mass casualty incidents. There is also a distinct shortage of theatre time and access to the intensive care unit. No simple, objective system has been described in the literature that assists with the daily triage and prioritisation of surgery for burns in low-resource setting on a continual basis. Mass casualty literature also describes burn assessment teams which can be used for specialized in-hospital triage, or specialized secondary triage, of patients who have already been admitted to a hospital. We therefore sought to utilise this concept and design and implement a simple, reproducible strategy that can be employed in similar settings where the health system is unable to meet the demand. This systematic approach aimed to select patients to be prioritised for early surgery based on a more urgent need and greater potential benefit for the patient.
This study aimed to describe the efficacy of this prioritisation system. The primary outcome was the ability to perform surgery in the priority group within three days of the decision. We also report the time from injury to operation and aimed to assess describe the outcomes of both the priority and non-priority groups. The study was not designed to compare outcomes between priority and non-priority groups.
We have moved the discussion around criteria used for priority decision and described reasons for criteria inclusion or exclusion.
The priority criteria were based on accepted burn care principles. For example, surgery is a cornerstone of sepsis management to eliminate the source and control ongoing contamination [16]. Where there is invasive burn wound infection or a high burden of non-viable tissue with risk of invasive wound infection, prompt surgery is warranted. In larger surface area deep burns, early surgery is considered part of physiological source control. Full thickness burns of the hand, foot or limb burn were selected when there was high risk of limb loss or significant deformity or dysfunction if not operated early. Obvious full thickness facial burns are typically associated with very large surface area injuries which in our setting are treated with palliative care. Typically, facial burns are partial thickness or mixed depth and managed expectantly and grafted at 3 weeks when demarcation is clear. Similarly with perineal burns. Most of our patients are children and young adults, with older patients being uncommon and generally having poor outcomes so were not included in priority criteria due to poor likely benefit and lack of geriatric specialists. Our selective strategy contrasts with the standard approach where the aim is to operate on all patients requiring surgery within 7 to 10 days of injury [3] on a first come first serve basis. We deliberately delayed transfer of non-priority patients from district level care to accommodate priority cases. Attempts were then made to provide surgery for the prioritised patients as soon as the decision for priority was made, by giving these patients preference on available burns lists and on the emergency surgery lists.
Comment 3
- The goal of the work is formulated quite specifically. In my opinion, the goal sounds a little unusual. As a rule, the goal (and the concept of treatment effectiveness) include the task of reducing mortality, reducing the incidence of complications, and similar clinical priorities. But the authors certainly have the right to set such a "technical" goal - reducing the duration of the preoperative period by expanding access to the operating room.
Response 3
Our goal was to achieve earlier access to surgery for the patients that would benefit most, acknowledging that in our setting, 5 out of 6 patients are unlikely to receive ‘early surgery’. We first aim to test this system’s efficacy in getting the selected group to theatre within our environment, and describe outcomes between the two groups. The study is not powered to test outcomes within this system, but we agree the ultimate goal of the surgeons work is to improve patient outcomes. We are currently writing up a second paper comparing patient outcomes in the system described in this paper with a different system that adopts a more conservative approach within a resource-constrained setting.
Comment 4
Comment Section "Materials and Methods". Line 107 - "Infection was marked present if systemic antimicrobials were 107 initiated". The meaning of the phrase is unclear. Did you think that a patient had a systemic infection if the doctor prescribed antibiotics to the patient? Is that right? Perhaps it means that "patients diagnosed with a systemic infection were prescribed antibiotics by the doctor"? Perhaps this misunderstanding is due to an inaccurate translation, but the phrase requires clarification.
In the Material and Methods section, it is absolutely necessary to describe how exactly the diagnosis of "sepsis" was established.
Response 4
Thank you for this comment, we agree that this aspect requires more detailed explanation. We have amended this section in the manuscript as follows:
Infection is a difficult diagnosis to make and was always made by the burn specialist surgeon in conjunction with an intensivist, and not merely by any doctor. Sepsis was always carefully considered, and diagnosis is based on clinical evaluation according to international guidelines, blood and Xray investigation and cultures. Antibiotics were initiated for suspected sepsis.
Comment 5
The Results section is written quite well. However, it remains unclear how the developed system of differentiated access to the operating room affected the clinical outcomes of treatment. Has the mortality rate among patients with severe burns become higher than it was before the introduction of the new system? Remained the same? Decreased? Perhaps this result (reduced complication rates and mortality) was not the goal of this work. But it is definitely the main purpose of the surgeon's work. Are there any data on this?
Response 5
This is an extremely important point. Our goal was to achieve earlier access to surgery in a specifically selected group of patients that we felt were most likely to benefit from early surgery, acknowledging that most patients would not have access to early operative management. The aim of the study was to determine whether early surgery could be achieved and in what proportion of patients. While we describe outcomes between the two groups, the study is not powered to provide meaningful analysis on this aspect, and we felt introducing historical data as a comparator was not appropriate, given the changes in system resources that have occurred recently in our setting (including reduced theatre lists and staffing).
We agree the goal of the surgeons work is to improve patient outcomes and we are currently writing up a second paper comparing patient outcomes in the system described in this paper with a different system.
Comment 6
- The Conclusions section begins with the phrase "This triage strategy represents a novel approach for access to burn surgery in a setting where global surgical standards are desirable but not always possible." I repeat, there are serious doubts that this is precisely a "novel approach." The criteria that the authors used to divide the patient flow are rather traditional. This, however, does not reduce the value of the presented article.
Response 6
Thank you for this comment: we agree the criteria we have applied are traditional, and not novel. Our aim was to provide a simple, objective and structured system for the selection of priority patients for early operative management, that could be applied in chronic overwhelmed environments. We believe the novelty is in providing such a structured system, which can then be improved upon and tested against more conservative approaches. We have reworded the conclusion to better reflect this.
Global surgery standards are desirable but not always possible. This paper describes a novel adaptation of the system for surgical access in a low resource setting faced with challenges in universal access to early surgery for all deep burns. We used a strategy of triage, previously described in mass casualty incidents to improve access to early surgery for the patients we believed would most benefit.
Comment 7
Thus, the presented article is very interesting, informative, relevant. At the same time, it requires some revision.
Response 7
Thank you for these comments: we have attempted to resolve these concerns.
Reviewer 3 Report
Comments and Suggestions for Authors
Relevant paper especially in low and moderate income countries.
Good design to triage high risk cases for operative treatment.
Methodology is sound and the results are well presented.
References are relevant and the conclusion is clear.
Area of minor revision - number of theatre trips for operation can be added and the logistics of complications relating to these episodes and whether any adherence to the current protocol has been maintained in multiple operations.
Author Response
General Comments
Relevant paper especially in low and moderate income countries.
Good design to triage high risk cases for operative treatment.
Methodology is sound and the results are well presented.
References are relevant and the conclusion is clear.
Response
Thank you for these comments.
Comment 1
Area of minor revision - number of theatre trips for operation can be added and the logistics of complications relating to these episodes and whether any adherence to the current protocol has been maintained in multiple operations.
Response 1
Thank you for these suggestions. This has been added to the results and marked in red.
If surgery was staged, adherence to the current protocol was maintained for all operations. Eleven patients had two operations, three patients had two operations and one patient had four operations. For simplicity we have presented data on first surgery only.
Round 2
Reviewer 2 Report
Comments and Suggestions for Authors
Dear colleagues!
I am quite satisfied with the answers to the questions I asked. The article, as I have already said, is very interesting; after the corrections, its structure has improved significantly.
The only remark I should mention is the disorder of the literary sources. The updated list of references contains 21 articles. However, the text contains references to 15 articles. Please, eliminate this misunderstanding.
Best regards
Author Response
Thank you. We are pleased the corrections are well received. Our apologies, we did not correct the references in the manuscript. This has now been done, and highlighted in red.
Reviewer 3 Report
Comments and Suggestions for Authors
satisfied with minor revision
Comments on the Quality of English Languagegood english representation
Author Response
Thank you, we are pleased the revision is acceptable.